# Epidemiology and burden of respiratory syncytial virus in Italian adults: A systematic review and meta-analysis

**Alexander Domnich**[1]*, **Giovanna Elisa Calabrò**[2]

**1** Hygiene Unit, San Martino Policlinico Hospital-IRCCS for Oncology and Neurosciences, Genoa, Italy,
**2** Section of Hygiene, Department of Life Sciences and Public Health, Università Cattolica del Sacro Cuore, Rome, Italy

* alexander.domnich@hsanmartino.it

**Data Availability Statement:** All raw data are within the manuscript and associated supporting materials.

**Funding:** The author(s) received no specific funding for this work.

## Abstract

### Objective

Respiratory syncytial virus (RSV) is a common respiratory pathogen not only in children, but also in adults. In view of a recent authorization of adult RSV vaccines in Italy, our research question was to quantify the epidemiology and burden of RSV in Italian adults.

### Methods

Observational studies on the epidemiology and clinical burden of laboratory-confirmed or record-coded RSV infection in Italian adults of any age were eligible. Studies with no separate data for Italian adults, modeling and other secondary publications were excluded. A literature search was performed in MEDLINE, Biological Abstracts, Global Health, Scopus and Web of Science on 22 November 2023. Critical appraisal was performed by means of a Joanna Briggs Institute checklist. Random-effects (RE) meta-analysis was performed to obtain pooled estimates and the observed heterogeneity was investigated by subgroup and meta-regression analyses. The protocol was prospectively registered (doi.org/10.17504/protocols.io.5qpvo32odv4o/v1).

### Results

Thirty-five studies were identified, most of which had at least one possible quality concern. RSV seasonal attack rates ranged from 0.8 ‰ in community-dwelling older adults to 10.9% in hematological outpatients. In the RE model, 4.5% (95% CI: 3.2–5.9%) of respiratory samples tested positive for RSV. This positivity prevalence was higher in older adults (4.4%) than in working-age adults (3.5%) and in outpatient (4.9%) than inpatient (2.9%) settings. According to the meta-regression, study location and sample size were also significant predictors of RSV detection frequency. The pooled estimate of in-hospital mortality was as high as 7.2% (95% CI: 4.7–10.3%). Data on other indicators of the diseases burden, such as complication and hospitalization rates, were unavailable.

**Competing interests:** Alexander Domnich and Giovanna Elisa Calabrò, the only authors of the manuscript, declare no competing interests. This does not alter our adherence to PLOS ONE policies on sharing data and materials.

## Conclusion

RSV poses a measurable burden on Italian adults, especially those of older age and with some co-morbidities. However, several data on the natural history of RSV disease are missing and should be established by future large-scale studies.

## Introduction

Together with seasonal influenza, respiratory syncytial virus (RSV) is a leading cause of respiratory infections and is responsible for a significant socioeconomic burden in all age groups, especially at the extremes of age [1, 2]. A recent modelling study [3] has estimated that on average, a total of 158,229 RSV-associated hospitalizations among European adults occur annually and 92% of these concerns the elderly. Contrary to young children, mortality attributable to RSV increased in both working-age and older adults [4].

Several systematic reviews and meta-analyses (SRMAs) [5–10] have investigated global epidemiology and burden of RSV in (older) adults. These reviews have advocated high incidence, hospitalization, mortality and case-fatality rates of RSV [5–10], which seem similar to seasonal influenza [9], and pointed out a substantial case under-ascertainment [10]. Moreover, there was a large between-country variation in estimates of the burden of disease (BoD) [6], which is driven by numerous factors, from climatic conditions [11] to the features of surveillance systems in place [12]. Notably, the available SRMAs [5–10] have identified only up to six studies conducted in Italy.

Two vaccines have been recently authorized to prevent lower respiratory tract disease (LRTD) caused by RSV in adults aged ≥60 years [13]. This age indication will be likely extended to younger individuals in the upcoming years. Some countries have already established RSV immunization policies. In the United States (US), a single dose of RSV vaccine is recommended to adults ≥60 years, as a part of shared clinical decision-making between patient and healthcare provider [14]. In the United Kingdom (UK), RSV vaccination is currently advised for older adults aged ≥75 years, being the most cost-effective option [15]. By contrast, as of January 2024, no recommendations have been issued in Italy.

Understanding country-specific BoD is a key driver for policy decisions on the introduction of new vaccines [16]. A systematic appraisal of the burden of RSV enables policy makers, health professionals and other relevant stakeholders to make informed decisions regarding the recently available vaccines. In this regard, SRMAs on different BoD indicators are important in the description of spatiotemporal distribution and variations between population subgroups potentially targeted by the novel preventive measures [17, 18]. Systematically appraised country-specific BoD indicators are also essential for all types of pharmacoeconomic models.

In Italy, a recent review [19] has assessed RSV BoD in pediatric outpatients, also with the aim to inform decision makers on the recent availability of a monoclonal antibody for RSV prevention in neonates. By including six studies, the authors found that 18–41% of children with respiratory infections were positive for RSV. Conversely, no reviews have systematically assessed burden of RSV in Italian adults. Indeed, RSV epidemiology is highly age-dependent, which hinders transferability of pediatric estimates to older populations. Furthermore, as we mentioned earlier, the available global-level reviews [5–10] were able to identify only a limited number of Italian studies. In this SRMA, we aimed to comprehensively collect and analyze available data on RSV epidemiology and BoD in Italian adults with the ultimate goal of informing and supporting National and local decision makers on the planification and implementation of vaccination strategies. In particular, we formulated the following research

question: "In a typical winter season, how many Italian adults contract RSV and how many of them develop complications, being hospitalized and die?".

## Materials and methods

### Reporting standards and protocol

PRISMA (preferred reporting items for SRMAs) statement [20] was adopted as a reporting standard (S1 Table). Methodological guidance for systematic reviews of observational epidemiological studies reporting prevalence and cumulative incidence data developed by the Joanna Briggs Institute (JBI) [21] was also consulted. The study protocol was prospectively registered with protocols.io [22] and no amendments to the original protocol were made.

### Eligibility criteria

All types of observational studies and published in any modality (e.g., peer-reviewed article, preprint, conference abstract, etc.) were potentially eligible. The CoCoPop (condition, context, and population) approach [21] was used to formulate the inclusion criteria. In particular, the condition of interest was RSV infection detected by any laboratory technique, including reverse-transcription polymerase chain reaction (RT-PCR), culture, immunofluorescence assay (IFA) and rapid antigen tests. Moreover, RSV-specific International Statistical Classification of Diseases and Related Health Problems (ICD) diagnosis codes (see below) were also considered a good proxy for the true RSV infection, since their specificity is as high as 99.6–99.8% [23, 24]. For the context, we considered studies conducted in Italy, in any setting (outpatient, inpatient or mixed), time and calendar period. Population consisted of adults defined as individuals aged ≥14 years, independently of their health conditions. Any reason or clinical entity [e.g., influenza-like illness (ILI), acute respiratory infection (ARI), severe ARI (SARI), clinical request for differential diagnosis] that triggers collection of biological samples (upper respiratory tract specimens including naso/oropharyngeal and nasal swabs; lower respiratory tract specimens including sputum, bronchoaspirates and bronchoalveolar lavage fluids) was eligible.

The following were set as exclusion criteria: (i) modeling, pharmacoeconomic and similar studies with no original data; (ii) insufficient data on RSV; (iii) studies on general population with no separate data for adults; (iv) multi-country studies with no separate data for Italy; (v) redundant publications.

### Study endpoints

RSV attack rate (cumulative incidence) was defined as the occurrence of laboratory-confirmed RSV detection in a population (symptomatic, asymptomatic or both) and in a specific period. RSV positivity prevalence was defined as proportion of RSV detections to the number of subjects tested. Prevalence of viral co-detections was described as number of samples tested positive for both RSV and any other respiratory virus to the total number of RSV-positive samples. Case-complication rate was defined as proportion of subjects who tested positive for RSV and developed ≥1 respiratory or extra-respiratory complication, such as pneumonia, exacerbation of chronic obstructive pulmonary disease (COPD), asthma, congestive heart failure, and other. As for drug use indicators, we considered frequency of antibiotic prescriptions among RSV-positive subjects. For what concerns inpatient outcomes, crude hospitalization, case-hospitalization (i.e., proportion of RSV-positive individuals who were hospitalized) rates, length of stay [mean with standard deviation (SD) or median with interquartile range (IQR)] and frequency of admission to intensive care units (ICUs) were of interest. Analogously, crude, in-hospital,

30-day mortality and case-fatality rates were eligible. For the indicators based on hospital discharge records (HDRs) or death certificates, we considered only RSV-specific codes, namely RSV pneumonia (ICD-9: 480.1; ICD-10: J12.1), acute bronchiolitis due to RSV (ICD-9: 466.11; ICD-10: J21.0), acute bronchitis due to RSV (ICD-10: J20.5), and RSV as the cause of diseases classified elsewhere (ICD-9: 079.6; ICD-10: B97.4).

When possible, all study endpoints were described overall, by age-group (working-age and older adults), RSV subtype (A and B) and season.

## Search strategy

The automatic search was performed on 22 November 2023 in the following databases: (i) MEDLINE via Ovid; (ii) Biological Abstracts via Ovid; (iii) Global Health via Ovid; (iv) Scopus and (v) Web of Science. In order to increase sensitivity, no filters or other restrictions (e.g., language or publication year) were applied. The search script considered both MeSH (medical subject headings) and text-wide terms and is reported in S2 Table.

We then performed a manual search through several modalities. First, the reference lists of the available global-level SRMAs [5–10] were checked. Second, a backward cross-reference checking of the included studies was carried out. Third, a forward citation search by using Google Scholar (https://scholar.google.com/) was conducted, as this search engine is better suited for identifying grey literature sources [25]. Fourth, periodic reports of the Italian surveillance reports on influenza and other respiratory viruses (https://respivirnet.iss.it) were examined. Finally, we screened abstract books and proceedings of some relevant conferences, including ECCMID (European Society of Clinical Microbiology and Infectious Diseases), ESWI (European Scientific Working Group on Influenza) and ReSViNET.

## Study selection

To manage references and remove duplicates, Zotero v.6.0.9 (Corporation for Digital Scholarship; Vienna, VA, USA) and Microsoft Excel (Microsoft Corporation; Seattle, WA, USA) were used. The resulting list of unique records underwent screening by assessing titles and/or abstracts and clearly irrelevant citations were discarded. Full texts of potentially pertinent publications were then downloaded and assessed for the above-described eligibility criteria. Study selection was finalized by performing the manual search, as described earlier. The entire process of study selection was performed by both reviewers, each working independently; eventual disagreements were solved by consensus.

## Data extraction and abstraction

The following data were extracted: (i) full citation record; (ii) study location; (iii) study period; (iv) study design; (v) study setting; (vi) main characteristics of the study population; (vii) sample size; (viii) funding source; (ix) eligibility criteria and case definitions; (x) methods used for case ascertainment; (xi) numerators and denominators used to compute the endpoints of interest described above; (xii) other potentially relevant information.

On the basis of period, studies were dichotomized on whether they overlapped with the COVID-19 pandemic, which had a significant impact on the circulation of RSV and other respiratory viruses. In particular, the northern hemisphere winter season 2020/2021 was characterized by a very limited circulation of RSV [19, 26]. Estimates for that season were extracted, but not included in the quantitative synthesis. Starting from the 2021/2022 season, RSV returned to the epidemiological scene [26] and therefore estimates from 2021/2022 onwards were fully considered. Multi-season studies that reported separate seasonal data were considered as distinct estimates [27]. Moreover, we distinguished between year-around studies and those conducted

during a typical RSV season, in which the probability of RSV detection is much higher [28]. We defined RSV epidemic season as a period between October and April [26].

On the basis of sample size, studies were median split. Regions of were categorized into three macro-areas of North (Aosta Valley, Liguria, Lombardy, Piedmont, Emilia-Romagna, Friuli-Venezia Giulia, Trentino-South Tyrol, Veneto), Center (Lazio, Marche, Tuscany, Umbria) and South (Abruzzo, Apulia, Basilicata, Calabria, Campania, Molise, Sicily, Sardinia).

Missing, unclear or presented only in graphical form data on relevant numerators and/or denominators were handled as follows. First, the corresponding author was contacted for clarification. In case of no reply, these data were imputed from the available percentages and/or by extracting data from figures using the WebPlotDigitizer v.4.6 software (https://automeris.io/WebPlotDigitizer). Data were extracted by AD and then validated by GEC.

## Critical appraisal

The JBI checklist for prevalence studies [21] was used to assess quality of the included studies. It was assessed independently by both reviewers and eventual conflicts were solved by consensus. Item 9 of the JBI checklist on the response rate was judged irrelevant for this study. Owing to a limited information available, risk of bias of conference abstracts, letters to the editor, short communications and similar was not assessed.

## Data synthesis

Tabulated data were first reassumed qualitatively and by visualizing forest plots. For quantitative synthesis, a proportional meta-analysis was undertaken according to the available recommendations [18, 21]. As heterogeneity was expected to be high, random-effects (RE) models with double arcsine transformation to stabilize variances were used. Pooled estimates were expressed as proportions with 95% Clopper-Pearson confidence intervals (CIs). Heterogeneity was quantified by means of the $\tau^2$ and $I^2$ statistics. Notably, high $I^2$ values do not necessarily mean that the data are inconsistent, as true heterogeneity is expected in prevalence estimates due to spatiotemporal differences [18]. The 95% prediction intervals (PIs) were also computed. As recommended, publication bias was not formally assessed, since there is no consensus about what a positive result in meta-analyses of proportions is [18].

To investigate the sources of heterogeneity across studies, both subgroup and meta-regression analyses were performed. In particular, the pre-specified subgroup analysis was performed by age-group (working-age and older adults), setting (outpatient, inpatient and mixed) and study period in relation to the COVID-19 pandemic (before the 2020/2021 season and after the 2020/2021 season). The meta-regression modeling was then conducted to examine the influence of study characteristics on the RSV-related endpoints. This latter was performed only when ≥10 estimates were available [29]. A leave-one-out sensitivity analysis was finally conducted to check the robustness of pooled estimates.

Meta-analysis was performed in R (R Foundation for Statistical Computing; Vienna, Austria) package "Meta" v. 6.5–0.

## Results

### Characteristics of the included studies

The automatic search generated 312 records, of which 171 were duplicates. Following screening of 141 records, 37 were judged potentially eligible. Twelve studies were excluded with reasons and are reported in S3 Table. Manual search identified further 12 studies and therefore the final list was composed of 35 studies corresponding to 37 publications [30–66]. Notably,

results relative to a retrospective cohort (henceforth referred to as "Boattini 2021–2023") were presented in three different publications [64–66]. Following correspondence with the corresponding author, who provided additional data, it was decided to include all these records. Moreover, relevant data (not reported within the article) were also obtained from other two corresponding authors [55, 59]. The entire study selection process is reported in Fig 1.

Most (83%; 29/35) studies included were full-length articles, while the remaining six were conference abstracts [45, 47], letters to editor [43, 57] and short papers [31, 39]. Principal characteristics of the studies are summarized in Table 1. Briefly, the studies covered a period from the 2001/2002 season to the 2022/2023 season and most (63%; 22/35) were conducted in the north of Italy. Approximately half (49%; 17/35) of the studies investigated more than one RSV season and the period of nine (26%) studies overlapped with the COVID-19 pandemic. Design of the majority of studies was judged cross-sectional (54%; 19/35) or surveillance (29%; 10/35) and the median sample size was 328 (range 43–28,500) patients. The study population was composed of outpatients, inpatients and mixed groups in 20% (7/35), 37% (13/35) and 31% (11/35) of studies respectively; the remaining four (11%) studies were focused on hematological [43–45] and cystic fibrosis [57] in-/outpatients. The setting of these latter four studies was therefore categorized as "immunocompromised patients". All but one studies used at least one laboratory assay (mostly RT-PCR) for RSV detection. The remaining study [35] analyzed RSV-specific HDRs. Within laboratory-based studies (n = 34), there was a high level of heterogeneity in terms of clinical entity

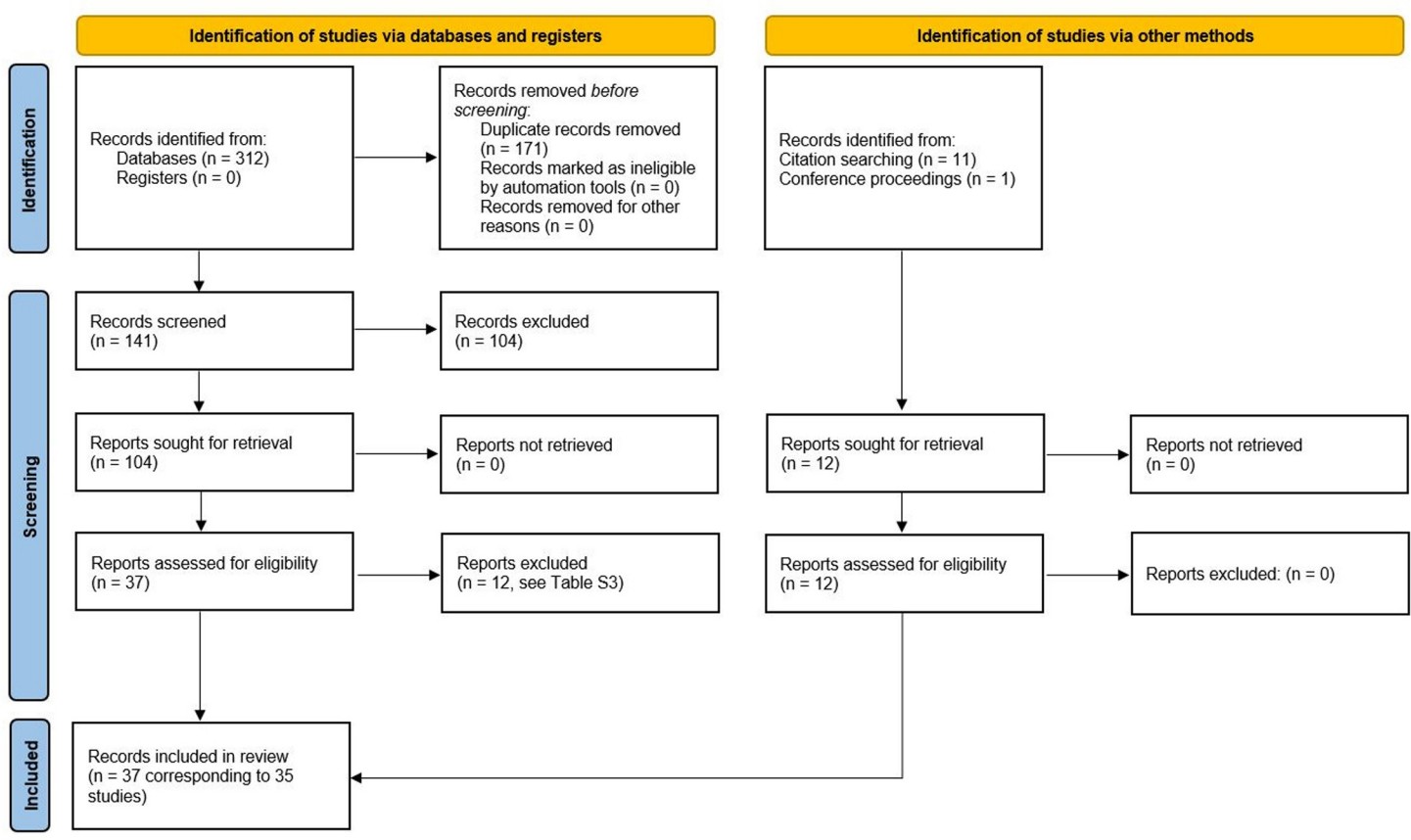

**Fig 1. PRISMA flow diagram of the study selection process.**

**Table 1. Characteristics of the studies included.**

| Study [Ref] | Study period | Design | Region (area) | Age, years[a] | Setting | Biological sample trigger or testing reason | Sample size[a] | Case ascertainment (specimen type) | Out-of-season samples | Funding |
|---|---|---|---|---|---|---|---|---|---|---|
| Rovida 2005 [30] | 10/2001–05/2002 | Cross-sectional | Lombardy (North) | ≥16 | Inpatient | ARI | 73 | RT-PCR and monoclonal antibodies (U) | Yes | Public |
| Minosse 2008 [31] | 04/2004–05/2005 | Surveillance | Lazio (Center) | ≥18 | Inpatient | Respiratory diseases (ARI, COPD, pneumonia) | 433 | RT-PCR (L) | Yes | Public |
| Puzzelli 2009 [32] | 11/2004–04/2007 | Surveillance | Lazio (Center), Campania (South) | ≥14 | Outpatient | ILI | 356 | RT-PCR (U) | No | Public |
| Costa 2007 [33] | 05/2005–10/2005 | Cross-sectional | Piedmont (North) | ≥17 | Mixed | Clinical request | 46 | Culture (L) | Yes | NA |
| Gerna 2009 [34] | 11/2006–05/2007 | Cross-sectional | Lombardy (North) | ≥18 | Mixed | ARI | 244 | RT-PCR (U/L) | Yes | Public |
| Cocchio 2023 [35] | 01/2007–12/2021 | Cross-sectional | Veneto (North) | ≥50 [b] | Inpatient | – | 250 | RSV-specific HDRs (NA) | – | Public |
| Gambarino 2009 [36] | Presumably 2008 | Cross-sectional | Piedmont (North) | ≥16 | Inpatient | Respiratory conditions | 324 | RT-PCR (L) | Yes | Public |
| Paba 2014 [37] | 02/2009–05/2011 | Validation of an assay | Lazio (Center) | ≥16 | Mixed | ILI | 178 | RT-PCR (U/L) | Yes | Public |
| Pierangeli 2011 [38] | 02/2009–03/2010 | Cross-sectional | Lazio (Center) | ≥18 | Mixed | Fever at ED admission or in the preceding 5 days and ≥1 respiratory diagnosis (ICD-9: 462, 466.0, 485, 480–486, 786.0, 786.2, 786.5, 793.1, 487) | 238 | RT-PCR (U) | No | Public |
| Nisii 2010 [39] | 05/2009–12/2009 | Cross-sectional | Lazio (Center) | ≥16 | Mixed | ILI | 544 | RT-PCR (U) | Yes | NA |
| Tramuto 2016 [40] | 07/2009–12/2012 | Cross-sectional | Sicily (South) | ≥15 | Inpatient | SARI with ICU admission | 192 | RT-PCR (U/L) | Yes | Public |
| Piralla 2017 [41] | 11/2009–04/2016 | Cross-sectional | Lombardy (North) | ≥18 | Inpatient | CAP with ICU admission | 376 | RT-PCR (U/L) | No | Public |
| Ansaldi 2012 [42] | 11/2010–04/2011 | Prospective cohort | Liguria (North) | ≥60 | Outpatient | ILI | 2551 (45 tested for RSV) | RT-PCR (U) | No | NA |
| Bigliardi 2015 [43] | 2010–2014 | Cross-sectional | Emilia-Romagna (North) | ≥15 | Immuno-compromised patients [c] | LRTD | 144 | Probably RT-PCR (L) | Yes | Public |
| Mikulska 2014 [44] | 01/2011–03/2011 | Prospective cohort | Liguria (North) | ≥18 | Immuno-compromised patients [c] | ILI, ARI, any new symptom and screening | 193 | RT-PCR (U/L) | No | Public |
| Passi 2019 [45] | 01/2011–03/2019 | Surveillance | Lombardy (North) | ≥18 | Immuno-compromised patients [c] | URTI | 151 | RT-PCR (U) | Yes | NA |
| Pellegrinelli 2020 [46] | 11/2014–04/2018 | Surveillance | Lombardy (North) | ≥16 | Outpatient | ILI | 706 | RT-PCR (U) | No | Public |
| Costa 2023 [47] | 11/2014–04/2022 | Cross-sectional | Liguria (North) | ≥18 | Mixed | Clinical request | 11,658 | RT-PCR (U/L) | No | Public |
| Tramuto 2021 [48] | 10/2015–04/2020 | Cross-sectional | Sicily (South) | >18 | Mixed | ILI (outpatients), SARI (inpatients) | 3727 | RT-PCR (U/L) | No | Private |
| Leli 2021 [49] | 01/2016–06/2020 | Cross-sectional | Piedmont (North) | ≥18 | Mixed | Respiratory infection | 375 | RT-PCR (U) | Yes | Public |
| Ciotti 2020 [50] | 10/2016–03/2019 | Cross-sectional | Lazio (Center) | ≥18 | Inpatient | Suspected or documented respiratory virus infection | 539 | RT-PCR (U/L) | Yes | Public |

(*Continued*)

**Table 1.** (Continued)

| Study [Ref] | Study period | Design | Region (area) | Age, years[a] | Setting | Biological sample trigger or testing reason | Sample size[a] | Case ascertainment (specimen type) | Out-of-season samples | Funding |
|---|---|---|---|---|---|---|---|---|---|---|
| De Francesco 2021 [51] | 01/2017–05/2021 | Cross-sectional | Lombardy (North) | ≥18 | Inpatient | SARI | 3974 | RT-PCR (U/L) | Yes | Public |
| Galli 2020 [52] | 11/2018–04/2019 | Validation of swab self-sampling | Lombardy (North) | ≥18 | Outpatient | ILI | 262 | RT-PCR (U) | No | Public |
| Domnich 2024 [53] | 11/2018–03/2020 | Surveillance | Liguria (North) | ≥18 | Outpatient | ILI | 1240 | RT-PCR (U) | No | Public and private |
| Spagnolello 2021 [54] | 01/2019–02/2019 | Cross-sectional | Lazio (Center) | >18 | Inpatient | CAP with ED admission | 75 | RT-PCR (U) | No | NA |
| Sberna 2022 [55] | 10/2019–12/2021 | Cross-sectional | Lazio (Center) | ≥18 | Inpatient | SARI | 557 | RT-PCR (U) | No | Public |
| Pierangeli 2022 [56] | 12/2018–04/2019 | Surveillance | Lombardy, South Tyrol (North), Veneto, Marche, Lazio (Center) [d] | ≥16 | Mixed | ILI (Lombardy) or clinical request (other regions) | Unclear [e] | RT-PCR (U/L) | No | Public |
| Scagnolari 2020 [57] | 11/2019–03/2020 | Retrospective cohort | Lazio (Center) | >18 | Immuno-compromised patients [f] | Medically attended respiratory disease or screening | 183 | RT-PCR (U) | No | Public |
| Galli 2021 [58] | 11/2019–04/2020 | Surveillance | Lombardy (North) | ≥15 | Outpatient | ILI | 401 | RT-PCR (U) | No | Public |
| Treggiari 2022 [59] | 11/2019–01/2022 | Cross-sectional | Veneto (North) | >18 | Mixed | Clinical request | ca 28,500 [g] | RT-PCR (U) | No | Public |
| Calderaro 2021 [60] | 12/2019–03/2020 | Surveillance | Emilia-Romagna (North) | ≥18 | Mixed | Clinical request | 332 | RT-PCR, IFA, culture (U/L) | No | Public |
| Milano 2023 [61] | 11/2021–04/2022 | Surveillance | Tuscany (Center) | ≥18 | Inpatient | SARI | 129 | RT-PCR (U) | No | Public and private |
| Panatto 2023 [62] | 12/2021–03/2022 | Surveillance | Liguria (North) | ≥18 | Outpatient | Any respiratory symptom | 1213 | RT-PCR (U) | No | Private |
| Santus 2023 [63] | 10/2022–03/2023 | Cross-sectional | Lombardy (North) | ≥18 | Inpatient | ILI at ED visit | 717 | RT-PCR (U) | No | Public |
| Boattini 2021–2023 [64–66] | 10/2017–04/2019 | Retrospective cohort | Piedmont (North) | ≥18 | Inpatient | LRTD | 43 | RT-PCR (U) | No | Public |

[a]In studies that enrolled also children, only adult individuals were considered.

[b]Age-group 5–49 years was excluded.

[c]All patients had hematological conditions.

[d]The study had also a center in Emilia-Romagna, which enrolled only young children 0–3 years.

[e]A total of 11,577 samples were tested for RSV and 423 adults tested positive, the total number of adults tested was not reported.

[f]All patients had cystic fibrosis.

[g]Number imputed from the available percentages.

ARI, Acute respiratory infection; CAP, community-acquired pneumonia; COPD, Chronic obstructive pulmonary disease; ED, Emergency department; HDR, Hospital discharge record; ICD, International Statistical Classification of Diseases and Related Health Problems; ICU, Intensive care unit; IFA, Immunofluorescence assay; ILI, Influenza-like illness; L, Lower respiratory tract specimens; LRTD, Lower respiratory tract disease; NA, Not available; RT-PCR, Reverse-transcription polymerase chain reaction; SARI, Severe Acute respiratory infection; U, Upper respiratory tract specimens; U/L, Upper and lower respiratory tract specimens; URTI, Upper respiratory tract infection.

triggering the specimen collection, especially in mixed and hospital settings. Six out 7 (86%) outpatient studies enrolled ILI patients. ILI and SARI were more frequently defined according to the European criteria (S4 Table). Out-of-season samples were frequent (38%; 13/34) and most studies analyzed only (56%; 19/34) upper respiratory tract specimens. The majority (74%; 26/35) of studies were funded by public institutions.

Fourteen [32, 38, 40, 41, 44, 46, 48, 51, 53, 58, 61–66] out of 29 full-length articles were judged at low risk of bias, while the remaining 15 studies [30, 33–37, 42, 49, 50, 52, 54–56, 59, 60, 64–66] had at least one possible source of bias (S5 Table). The most frequent criticalities concerned insufficient description of patients and setting, inappropriate sample frame and small sample size.

## RSV attack rate

Three cohort studies [42, 44, 57] allowed to determine RSV attack rates. The first one [42] consisted of a cohort of 2551 community-dwelling adults aged ≥60 years who were actively surveilled for ILI between November 2010 and April 2011. A total of 45 ILI cases were prospectively identified and these cases were tested in RT-PCR. Other 63 ILI cases were identified retrospectively and thus not tested in RT-PCR. Two out of 45 samples tested positive for RSV, giving a symptomatic attack rate of 0.8 ‰ (95% CI: 0.1–2.8‰). If corrected for undertesting, the cumulative incidence would rise to 1.9 ‰ [42]. Mikulska et al. [44] followed for three months (from January to March 2011) 193 hematological outpatients. A total of 21 swabs tested positive for RSV, with an overall attack rate of 10.9% (95% CI: 6.9–16.2%). Of RSV-positive subjects, 18 (85.7%) were symptomatic, while the remaining three (14.3%) patients were asymptomatic [44]. In a retrospective cohort of adult (>18 years) subjects affected by cystic fibrosis [57] eight out of 183 patients tested positive for RSV between November 2019 and March 2020, giving an attack rate of 4.4% (95% CI: 1.9–8.4%). Owing to different study populations, a pooled analysis of these three studies was judged unfeasible.

## Prevalence of RSV positivity

A total of 85 RSV positivity prevalence estimates were extracted from 32 studies, of which 42 (S6 Table), 21 (S7 Table) and 22 (S8 Table) concerned adults of any age, working-age and older adults, respectively. Seven estimates from three studies [51, 55, 59] covered exclusively the 2020/2021 season and were excluded from the pooled analysis.

As shown in Fig 2, the positivity rate in adults of any age ranged from 0.2% to 35.7%. The RE pooled estimate was 4.5% (95% CI: 3.2–5.9%). As expected, the heterogeneity was high ($I^2$ = 93.2%) and the 95% PI was 0.0–15.9%. Immunocompromised patients showed the highest RSV positivity prevalence of 11.5%. Compared with outpatients (4.9%), inpatients had lower prevalence of RSV (2.9%). Studies conducted in both settings showed an intermediate pooled estimate (3.7%). Omission of single studies did not alter significantly the observed pooled proportions.

In the subgroup analysis by age-group (Fig 3), working-age adults showed lower (3.5%) RSV prevalence than older adults (4.4%). RSV detection was more frequent in outpatients than inpatients in both age-specific RE models. RSV detection was also higher in studies conducted before the COVID-19 pandemic than among those performed during the pandemic (i.e., seasons 2021/2022 and 2022/2023) (S1 Fig).

A meta-regression analysis was conducted to investigate sources of the heterogeneity observed (S9 Table). Apart from the study setting, geographic area and sample size explained some variance in the reported proportions. In particular, compared with northern regions (5.5%), RSV prevalence was as twice as lower in central Italy (2.7%) (S2 Fig). Studies that enrolled <300 patients reported significantly higher prevalence (6.3%) estimates than larger

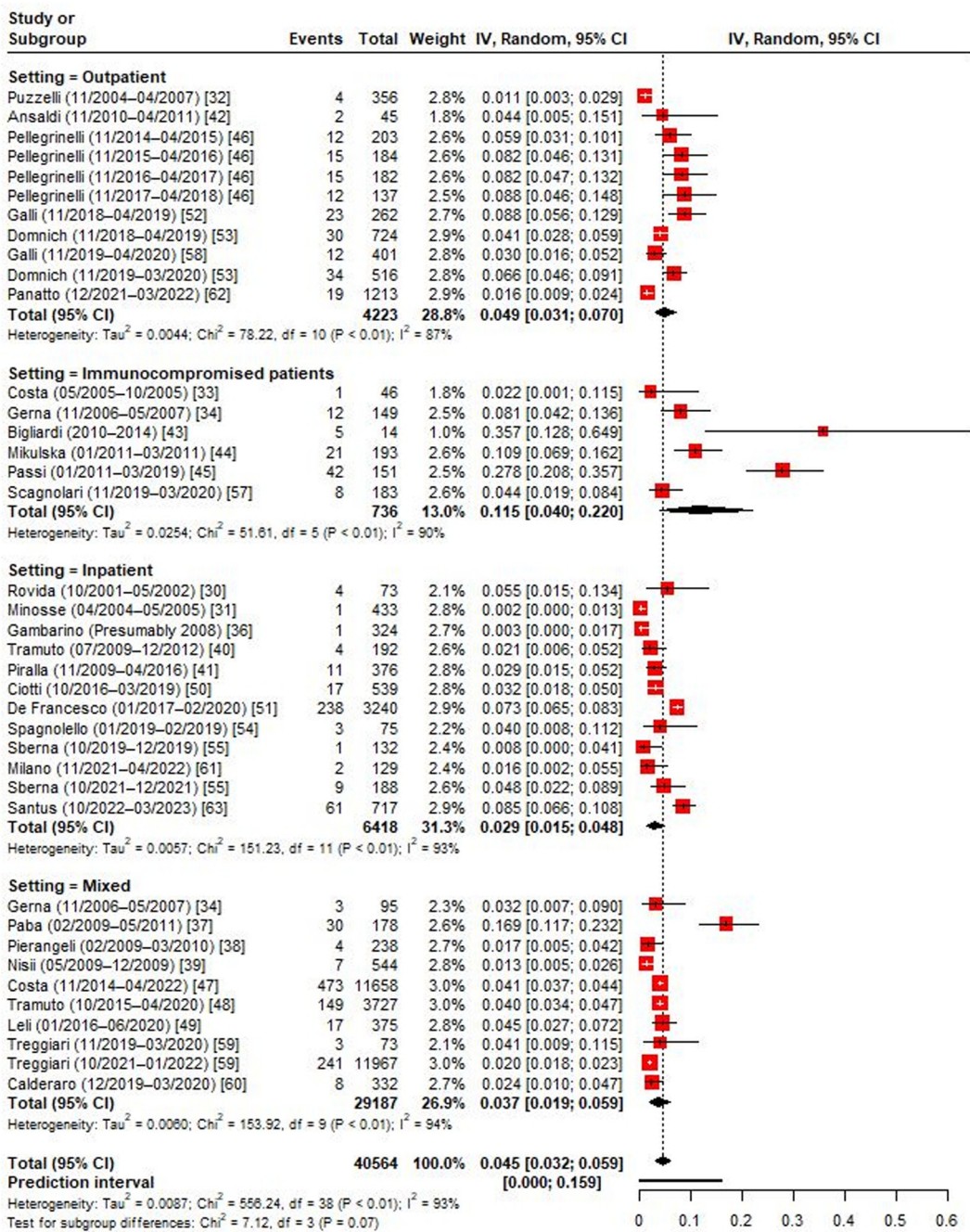

**Fig 2. Forest plot of the random-effects model on RSV positivity prevalence among Italian adults of any age, by setting.**

studies (3.0%) (S3 Fig). Publication year, public funding, number of seasons, inclusion of out-of-season samples, specimen type and risk of bias were not associated with the reported RSV detection rate (S9 Table).

Data on RSV subtypes was reported in nine studies [39, 40, 44, 46–48, 53, 56, 62]. In these studies, both subtypes co-circulated, although a relative dominance on one subtype over another varied by study period (S10 Table). On average, from 2009 to 2022 both subtypes were

**Working-age adults**

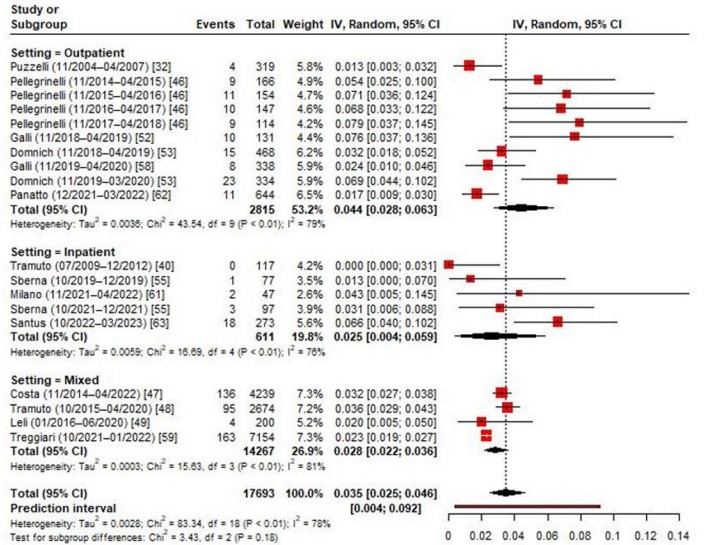

**Older adults**

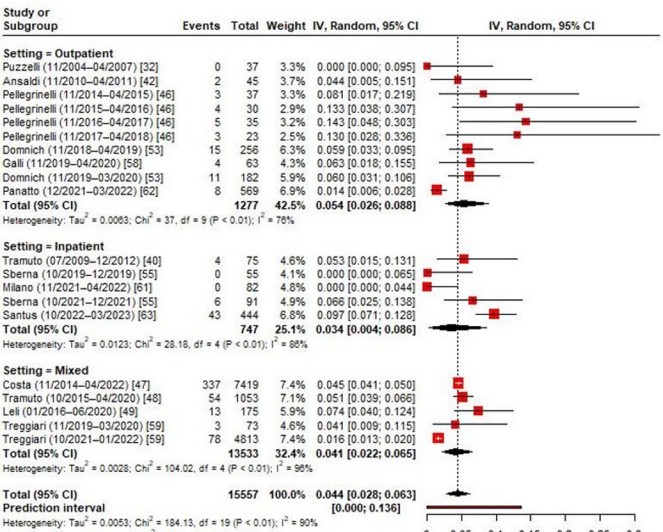

**Fig 3. Forest plot of the random-effects model on RSV positivity prevalence among Italian working-age and older adults, by setting.**

detected in almost equal proportions (RE model estimate for RSV B: 56.3%; 95% CI: 44.1–68.1%; $I^2$ = 88.4%) (S4 Fig).

Sanger sequencing was performed in three studies [46, 56, 62] conducted between 2014/2015 and 2021/2022 seasons. During the 2014/15 and 2015/16 seasons, both NA1 and ON1 genotypes of RSV A were circulating [46]. Conversely, starting from the 2017/2018 season all RSV A strains clustered within the ON1 genotype [46, 56, 62]. All RSV B strains belonged to the BA genotypes [46, 56, 62].

## Prevalence of viral co-detections

Thirteen studies reported data on viral co-detections [30, 31, 36, 37, 40, 47, 51, 53, 54, 60–63]. However, only five estimates [37, 47, 51, 53, 63] were based on a sufficient (≥30) number of RSV-positive samples and were therefore less prone to the small study effect (S11 Table). In these studies, 2.9–18.6% of RSV-positive specimens tested also positive for other respiratory viruses. In the RE model (S5 Fig), the pooled estimate was 8.6% (95% CI: 3.2–15.9%) and the heterogeneity was high ($I^2$ = 92.0%). Influenza A, seasonal coronaviruses, rhinovirus and para-influenza viruses were detected more frequently (S11 Table). These results should be interpreted cautiously, since the multiplex RT-PCR panels used differ from the point of view of antigens included.

## Severe RSV disease

The prespecified outcomes ascribable to severe RSV disease were reported in nine studies [35, 41, 42, 44, 50, 54, 56, 63, 64–66] (Table 2). Pierangeli et al. [56] reported that 63.1%, 63.3% and 79.3% of adults (both out- and inpatients) aged 16–65, 65–80 and >80 years had antibiotic pre-scriptions. The study, however, did not report raw data. In an outpatient cohort (n = 2551) of adults aged ≥60 years [42], none of seven cases of community-acquired pneumonia (CAP) tested positive for RSV. Ciotti et al. [50] reported that among 23 hospitalized cases of viral pneumonia, one (4.3%) was due to RSV. In a multi-season (from 2009/2010 to 2015/2016)

**Table 2. Severe RSV-related outcomes, by study.**

| Study | Age, years | Outcome | % (n/N) | 95% CI |
|---|---|---|---|---|
| Cocchio 2023 [35] | 50–69 | % in-hospital mortality | 4.9 (4/81) | 1.4–12.2 |
| | ≥70 | | 7.1 (12/169) | 3.7–12.1 |
| Piralla 2017 [41] | ≥18 | % RSV detection in patients with CAP admitted to ICU | 2.9 (11/376) | 1.5–5.2 |
| Ansaldi 2012 [42] | ≥60 | % RSV detection in patients with CAP | 0 (0/7) | 0–41.0 |
| Mikulska 2014 [44] | ≥18 | % RSV-positive hematological patients who developed LRTD | 10.5 (2/19) | 1.3–33.1 |
| | | % 30-day mortality | 0 (0/19) | 0–17.6 |
| Ciotti 2020 [50] | ≥18 | % RSV detection in patients hospitalized for viral pneumonia | 4.3 (1/23) | 0.1–21.9 |
| Spagnolello 2021 [54] | >18 | % RSV detection in ED patients with CAP | 4.0 (3/75) | 1.5–13.1 |
| Pierangeli 2022 [56] | 16–65 | % antibiotic use in RSV-positive subjects | 63.1 (NA) | NA |
| | 65–80 | | 63.3 (NA) | NA |
| | >80 | | 79.3 (NA) | NA |
| Santus 2023 [63] | ≥18 | % antibiotic use in RSV-positive inpatients | 78.7 (48/61) | 66.3–88.1 |
| | | Median length of hospital stay, days | 13 [a] | 7–22 [b] |
| | | % RSV-positive inpatients with acute respiratory failure, need for respiratory support, shock, sepsis, ICU admission or in-hospital death | 70.5 (43/61) | 57.4–81.5 |
| | | % RSV-positive inpatients with ICU admission | 6.6 (4/61) | 1.8–15.9 |
| | | % in-hospital mortality | 6.6 (4/61) | 1.8–15.9 |
| Boattini 2021–2023 [64–66] | ≥18 | % pneumonia in hospitalized RSV-positive subjects | 58.1 (25/43) | 42.1–73.0 |
| | ≥18 | Median length of hospital stay, days | 19 [a] | 11–30 [b] |
| | ≥65 | | 23 [a] | 8–34 [b] |
| | ≥18 | % invasive and non-invasive mechanical ventilation in hospitalized RSV-positive subjects | 30.2 (13/43) | 17.2–46.1 |
| | ≥65 | | 27.3 (3/11) | 6.0–61.0 |
| | ≥18 | % in-hospital mortality | 16.3 (7/43) | 6.8–30.7 |
| | ≥65 | | 9.1 (1/11) | 0.2–41.3 |

[a]Median.

[b]Interquartile range.

CAP, community-acquired pneumonia; CI, Confidence interval; ED, Emergency department; ICU, Intensive care unit; LRTD, Lower respiratory tract disease; NA, Not available.

study by Piralla et al. [41], the prevalence of RSV among subjects (11/376) with CAP admitted to ICU was 2.9%. Among 19 RSV-positive hematological patients [44], two (10.5%) developed LRTD and the 30-day mortality was 0% (95% CI: 0–17.6%). Spagnolello et al. [54] reported that among CAP patients who referred to emergency departments (n = 75), three tested positive for RSV (4.0%; 95% CI: 1.5–13.1%). However, the study period was limited to one month only (from 15 January to 22 February 2019) [54].

In a hospitalized cohort of RSV-positive adults (n = 61) [63], severe RSV disease (defined as presence of at least one of the following: acute respiratory failure, need for respiratory support, shock, sepsis, ICU admission or in-hospital death) was observed in 70.5% of patients. Antibiotics, systemic steroids, inhaled steroids, oseltamivir and other antivirals were administered to 79.0%, 65.6%, 70.0%, 13.1% and 8.2% of patients respectively. The median length of stay was 13 (IQR: 7–22) days. ICU admission and in-hospital mortality rates were both 6.6%.

In a retrospective cohort study by Boattini et al. [64–66], 58.1% (n = 43) of RSV-positive hospitalized adults ≥18 years had radiologically-confirmed pneumonia. The median length of stay for the entire cohort was 19 (IQR: 11–30) days, while it increased up to 23 (IQR: 8–34)

days for adults aged ≥65 years. A total of 30.2% of subjects needed invasive and non-invasive mechanical ventilation, while the in-hospital mortality was 16.3%.

Cocchio et al. [35] analyzed HDRs for RSV-specific ICD-9 codes in Veneto for 15 consecutive years (2007–2021). Of the total of 6961 (5818, 741 and 402 for acute bronchiolitis due to RSV, pneumonia due to RSV and RSV, respectively) hospitalizations, 81 (1.2%; 8, 41 and 32 for acute bronchiolitis due to RSV, pneumonia due to RSV and RSV, respectively) and 169 (2.4%; 31, 74 and 64 for acute bronchiolitis due to RSV, pneumonia due to RSV and RSV, respectively) encounters occurred in adults aged 50–69 and ≥70 years, respectively. The length of stay increased linearly with the increasing age for acute bronchiolitis due to RSV (β = 0.080; 95% CI: 0.059–0.100; P < 0.001), pneumonia due to RSV (β = 0.109; 95% CI: 0.083–0.135; P < 0.001) and RSV (β = 0.063; 95% CI: 0.045–0.081; P < 0.001), respectively. Of 23 deaths identified, 73.9% (17/23) were registered in subjects aged ≥50 years. The overall in-hospital mortality rate was 4.9% (4/81; 0%, 7.3% and 3.1% for acute bronchiolitis due to RSV, pneumonia due to RSV and RSV, respectively) and 7.1% (12/169; 12.9%, 10.8% and 0% for acute bronchiolitis due to RSV, pneumonia due to RSV and RSV, respectively) in subjects aged 50–69 and ≥70 years, respectively.

As reported in Table 2, only the outcome of in-hospital mortality was reported in more studies [35, 63–66]. In a pooled analysis of four estimates (S6 Fig), the in-hospital mortality was 7.2% (95% CI: 4.7–10.3%). The heterogeneity was relatively low ($I^2$ = 30.6%) and the 95% PI was 2.2–14.5%.

## Discussion

This SRMA showed that RSV is a frequent respiratory pathogen that poses a measurable burden to Italian adults. However, this BoD is unevenly distributed within the Italian adult population, being higher in older age groups and subjects with underlying health conditions. Our findings may inform principal stakeholders and support policy makers on the introduction of the recently authorized adults RSV vaccines [13] with the aim to ensure effective and coherent resource allocation. Analogously, our study is a good starting point for the upcoming cost-effectiveness and budget impact models. We finally identified principal data gaps, which must be addressed by future research.

Our meta-analysis showed that 4.5% of respiratory samples tests positive for RSV and the 95% PI suggested that the next estimate would be between 0% and 16%. This uncertainty reflects both natural between-season variation in the circulation of RSV and characteristics of the population surveilled. We indeed showed that compared with hospitalized individuals, community-dwelling adults had higher positivity prevalence (4.9% vs 2.9%). At the same time, compared with working-age adults (3.5%), RSV prevalence was higher in older adults (4.4%) and individuals with immunosuppressive disorders (11.5%). It should be stressed that all these pooled estimates are probably conservative, as a certain level of underestimation is likely for several reasons. First, most primary care studies were conducted within the existing influenza surveillance framework that usually relied on ILI-based case definitions. Different studies [53, 67, 68] have proved that a significant proportion of RSV-positive adults have no fever and therefore ILI and fever-based case definitions intrinsically underestimate the true incidence of RSV. Second, several hospital-based studies considered only one type of specimen. It has been, for example, shown [1] that adding RT-PCR of sputum to that of nasal or nasopharyngeal swab specimens increases RSV detection by 39–100%. Indeed, RSV may migrate from the upper to lower respiratory tracts and thus RT-PCR performed on naso/oropharyngeal swabs would produce a false-negative result [28].

When comparing our results to those of the global-level SRMAs [5–8], we noted both similarities and differences. Nguyen-Van-Tam et al. [7] have reported that the cumulative attack rate of RSV in older adults ranged from 0.27‰ to 108‰. This 400-fold difference [7] is unlikely to be explained by the epidemiology of RSV alone, but is rather linked to the design of single studies. As for RSV incidence in industrialized countries, Shi et al. obtained pooled estimates of 6.7‰ (95% CI: 1.4–31.5‰) [5] and 30.3‰ (95% CI: 5.3–59.9‰) [8] for older and high-risk adults, respectively. In Italy, the available incidence estimates were 1.9 ‰ [42] and 4.4–10.9% [44, 57] for older adults and adults with immunosuppressive conditions, respectively. Italian data, however, came from single-season studies and thus affected by RSV circulation patterns in those particular seasons. RSV positivity prevalence appears instead more consistent. For instance, a meta-analysis of European studies by Tin Tin Htar et al. [6] reported that 7% (95% CI: 4–11%), 9% (95% CI: 4–17%) and 10% (95% CI: 5–16%) of ILI/ARI cases in all adults, subjects aged <50 and ≥50 years, respectively, tested positive for RSV. Notably, that SRMA [6] included only one Italian study and reported that only 1% of ILI cases in Italy were positive for RSV. By analyzing RSV prevalence estimates from 32 studies, we were able to obtain higher positivity rates, which were more aligned with global-level figures. For instance, Shi et al. [5] estimated that RSV positivity among hospitalized older adults with ARI was 4.4% (95% CI: 3.0–6.5%). Our meta-analytical proportions were in line with these figures and the 95% CIs largely overlapped. For what concerns mortality indicators, the pooled in-hospital mortality in Italy was 7.2%. In comparison, the case-fatality proportions ranged from 1.6% (95% CI: 0.7–3.8%) [5] to 8.2% (95% CI 5.5–11.9%) [7] in the available global-level SRMAs.

Apart from age, underlying immunosuppressive conditions and setting, RSV positivity prevalence was associated with the study location and sample size. Studies conducted in the north of Italy reported on average a higher frequency of RSV. Compared with central and southern regions, mean winter temperature in northern regions is lower [69]. Lower ambient temperature is a well-known positive predictor of RSV detection frequency [70–72]. Smaller studies tended to report a higher RSV prevalence, which is consistent with the small study effect. Fortunately, the double arcsine transformation prevents the undue large weights for these studies [73]. Future epidemiological studies should *a priori* determine an adequate number of subjects to be tested for RSV.

Despite a relatively high number of publications identified, most studies were limited to quantifying prevalence of RSV following clinical request or in the context of influenza surveillance and were not specifically designed to investigate RSV epidemiology and the associated BoD. Only a few papers went beyond mere positivity prevalence reports. Currently, no cohort studies have evaluated the natural history of RSV disease in Italian adults. In particular, it remains unclear how many RSV-positive adult outpatients develop respiratory and extra-respiratory complications and being (re)-hospitalized. Some of these data gaps could be filled by a retrospective analysis of HDRs, but only if matched with laboratory data. Indeed, many RSV-positive patients could be attributed alternative diagnostic codes. Despite being highly specific and useful for the analysis of case-fatality rates, resource use and temporal trends, studies of RSV-specific HDRs in adults inevitably lead to the underestimation of RSV BoD. Cai et al. [24] have documented that the RSV-specific ICD-10 codes had a very low sensitivity of 6%. Retrospective cohort studies based on HDRs only (i.e., without matching with laboratory data) would identify an implausibly low number of RSV-specific diagnostic codes [35, 74]. Another important data gap concerns RSV epidemiology and BoD in residents of long-term care facilities, who may be the primary target population for vaccination.

As we also mentioned earlier, the main limitation of the current evidence, which may bias RSV detection rates, include the fact that several studies were conducted within influenza surveillance framework and used ILI- or influenza-specific case definitions. Conversely, only few

studies used a more sensible ARI case definition. As a consequence, our pooled estimates for RSV positivity rates showed a considerable heterogeneity in terms $I^2$ statistics. However, it should be stressed that high $I^2$ values are extremely common in meta-analyses of prevalence and not always synonymous of true heterogeneity [75]. Indeed, spatiotemporal patterns of RSV activity displays a high-level true heterogeneity in terms of both seasonal and spatial variations [76]. In view of the availability of novel preventive measures, there is an urgent need to harmonize RSV case definition. Another important shortcoming of the available evidence is lack of studies on the natural history of an RSV episode and some important BoD indicators like the occurrence of respiratory and extra-respiratory complications and hospitalization rates. Although these data are of primary importance for robust modeling and cost-effectiveness analyses, these latter data are scant event at global level [5–10]. Hopefully, recently some established RSV-specific international surveillance networks will shed light on the consequences of RSV infection in adults [77].

At review level, we must acknowledge that some potentially relevant studies were not identified because their primary goal was not related to RSV and thus were indexed alternatively. Furthermore, although we performed literature search in several platforms, other relevant databases (e.g., EMBASE) were not searched, as we had no access to them. Finally, in the absence of a universally recognized RSV case definition, reviewer's judgement on the risk of bias due to sample frame is rather subjective. In the present appraisal, we intentionally did not downgrade studies that used ILI- or influenza-specific syndromic definitions. Although it is believed that ILI-based surveillance underestimates RSV detection rates [68], a plurennial study by Sáez-López et al. [67] has found no difference between ILI- and ARI-based RSV case definitions in terms of sensitivity and specificity.

The approval of the new vaccines depends on national authorities. This decision occurs within the framework of developing national immunization programs and necessitates the utilization of assessment tools, such as the health technology assessment (HTA) [78]. This process relies on a comprehensive understanding of the epidemiology, BoD and pertinent economic analyses. Furthermore, HTA organizations are also evaluating the implementation of guidelines that take into account, in the assessment of new technologies, also their societal value [79]. This objective aligns with the "Broader value of vaccines" framework proposed by Bell et al. [80]. The framework encompasses various vaccine effects, including strictly health-related impacts, focusing on the health of vaccinated individuals; general health effects, considering the influence of vaccination on the health of the unvaccinated population; economic effects on the healthcare system, involving the costs and corresponding budgetary compensations; and societal economic effects, evaluating broader economic of vaccines impacts such as effects on productivity and macroeconomic growth from a societal perspective.

Incorporating these diverse effects is crucial in the HTA processes [79].

In conclusion, this article provides a comprehensive assessment of the RSV epidemiology and its BoD in the Italian adult population. A systematic assessment of the full value of a new vaccination necessitates access to national data regarding the epidemiological burden of the disease and the presence of effective surveillance systems. Data plays a pivotal role in generating knowledge and evidence about diseases. Only through evidence can priority actions for the control of infectious diseases be discerned, facilitating the promotion of an informed decision-making process. This, in turn, supports the implementation of value-based immunization strategies. In this paper, we comprehensively assessed RSV epidemiology and BoD in the Italian adult population. Within its shortcomings, this SRMA supports the need to prioritize evaluation of the novel RSV vaccines by Italian decision makers, especially for older adults and those affected by immunosuppressive conditions.

## Supporting information

**S1 Fig. RSV positivity prevalence among Italian adults of any age, by study period in relation to the COVID-19 pandemic.**
(DOCX)

**S2 Fig. RSV positivity prevalence among Italian adults of any age, by study geographic area.**
(DOCX)

**S3 Fig. RSV positivity prevalence among Italian adults of any age, by study sample size.**
(DOCX)

**S4 Fig. Prevalence of RSV subtype B among Italian adults of any age (prevalence of RSV subtype A may be computed as 1 –prevalence of RSV B).**
(DOCX)

**S5 Fig. Frequency of viral co-detections among RSV-positive Italian adults of any age.**
(DOCX)

**S6 Fig. In-hospital mortality among RSV-positive Italian adults of any age.**
(DOCX)

**S1 Table. PRISMA (Preferred Reporting Items for Systematic reviews and Meta-Analyses) checklist.**
(DOCX)

**S2 Table. Algorithm for the automatic search, by citation database.**
(DOCX)

**S3 Table. Excluded studies with reasons.**
(DOCX)

**S4 Table. Syndromic definitions used the studies analyzed.**
(DOCX)

**S5 Table. Risk of bias of the studies included according to the Joanna Briggs Institute (JBI) checklist for prevalence/incidence studies.**
(DOCX)

**S6 Table. RSV positivity prevalence among Italian adults of any age, by setting.**
(DOCX)

**S7 Table. RSV positivity prevalence among Italian working-age adults, by setting.**
(DOCX)

**S8 Table. RSV positivity prevalence among Italian older adults, by setting.**
(DOCX)

**S9 Table. Meta-regression analysis to investigate sources of heterogeneity in RSV positivity prevalence among Italian adults of any age.**
(DOCX)

**S10 Table. RSV positivity prevalence among Italian adults, by subtype.**
(DOCX)

**S11 Table. Frequency of viral co-detections among RSV-positive Italian adults of any age.**
(DOCX)

## Author Contributions

**Conceptualization:** Alexander Domnich, Giovanna Elisa Calabrò.

**Data curation:** Alexander Domnich, Giovanna Elisa Calabrò.

**Formal analysis:** Alexander Domnich, Giovanna Elisa Calabrò.

**Investigation:** Alexander Domnich, Giovanna Elisa Calabrò.

**Methodology:** Alexander Domnich, Giovanna Elisa Calabrò.

**Software:** Alexander Domnich, Giovanna Elisa Calabrò.

**Validation:** Alexander Domnich, Giovanna Elisa Calabrò.

**Visualization:** Alexander Domnich, Giovanna Elisa Calabrò.

**Writing – original draft:** Alexander Domnich, Giovanna Elisa Calabrò.

**Writing – review & editing:** Alexander Domnich, Giovanna Elisa Calabrò.

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
