## [Decision Letter · Decision Letter 0]

4 Feb 2024

PONE-D-24-00880Epidemiology and burden of respiratory syncytial virus in Italian adults: A systematic review and meta-analysisPLOS ONE

Dear Dr. Domnich,

Thank you for submitting your manuscript to PLOS ONE. After careful consideration, we feel that it has merit but does not fully meet PLOS ONE’s publication criteria as it currently stands. Therefore, we invite you to submit a revised version of the manuscript that addresses the points raised during the review process.

**ACADEMIC EDITOR: **Please address all reviewers' comments which you can find below. 

We look forward to receiving your revised manuscript.

Kind regards,

Victor Daniel Miron

Academic Editor

PLOS ONE

Journal Requirements:

"AD and GEC declare no competing interests."

3. We note that this manuscript is a systematic review or meta-analysis; our author guidelines therefore require that you use PRISMA guidance to help improve reporting quality of this type of study. Please upload copies of the completed PRISMA checklist as Supporting Information with a file name “PRISMA checklist”.

**Reviewers' comments**:

Reviewer's Responses to Questions

**Comments to the Author**

1. Is the manuscript technically sound, and do the data support the conclusions?

Reviewer #1: Yes

Reviewer #2: Yes

Reviewer #3: Partly

Reviewer #4: Partly

2. Has the statistical analysis been performed appropriately and rigorously? 

Reviewer #1: Yes

Reviewer #2: Yes

Reviewer #3: Yes

Reviewer #4: Yes

3. Have the authors made all data underlying the findings in their manuscript fully available?

Reviewer #1: Yes

Reviewer #2: Yes

Reviewer #3: Yes

Reviewer #4: Yes

4. Is the manuscript presented in an intelligible fashion and written in standard English?

Reviewer #1: Yes

Reviewer #2: Yes

Reviewer #3: Yes

Reviewer #4: Yes

5. Review Comments to the Author

Reviewer #1: This is a comprehensive assessment of the respiratory burden of RSV in italian adults. The methodology is robust. The article is well written.

I would add in the discussion and in the title that this work focused on the respiratory burden of RSV (and not the well-known cardio-vascular complications or other middle/long term complications such as re hospitalizations, dependancy etc...)

Reviewer #2: Authors of this manuscript looked into epidemiology and burden of RSV infection in Italian adults. Their research was carried our as a systematic literature review and met analysis. Overall, this works shows a very clear structure and follows the generally accepted rules for such research. The field of RSV epidemiological studies is pretty broad and with varying data and output quality. It is a strength of this work that authors focused on relevant outcomes and parameters to be extracted from analysed research. Data is well described and conclusions are well supported by their findings. As the authors state, this article provides a comprehensive assessment of the RSV epidemiology and BOD in the Italian population. Shortcomings of the existing literature are in a way limiting the strength of the conclusions of the paper, which is - of course - not a fault of the submitting researcher.

Reviewer #3: Dear all, The systematic review presented is relevant and has important contributions to the literature; however, some items must be reviewed when conducting the study. I suggest checking Prisma and faithfully following the checklist established as a guide for quality systematic reviews.

Page MJ, McKenzie JE, Bossuyt PM, Boutron I, Hoffmann TC, Mulrow CD, et al. The PRISMA 2020 statement: an updated guideline for reporting systematic reviews. BMJ 2021;372:n71. doi: 10.1136/bmj.n71

In the abstract:

- There is no clear statement of the main research question.

- There is no specification of inclusion and exclusion criteria, nor are the databases used to identify the studies described.

- There is no description of the last date on which the literature searches were carried out.

- There is no detailed description of the methods used to assess the risk of bias in the included studies. Authors must include the registration number of the review protocol in the study abstract.

- There is no adequate synthesis of the main results.

- The conclusion of the abstract talks about public policies, assessment of health technologies and vaccines. This is not the conclusion of a systematic review. Nor is the main finding of this study added to a pertinent conclusion related to what the results of the study contribute to the literature based on their possible generalizations.

In the article:

The authors claim to use Prisma and follow the guidelines of Joanna Briggs' institute; however, important details highlighted by these standards need to be added and clearly described, allowing for replicability of the review and transparency in its methodological conduct.

The authors use the acronym CoCoPop (condition, context, and population) which is usually used for scoping reviews. For cross-sectional studies, the acronym PECO (pollution, exposure, comparison, and outcome) would be ideal.

What is the guiding question of the study? What are the prevalence, incidence, and risk factors of respiratory syncytial virus in Italian adults? This is not clear.

The eligibility criteria established by the authors are quite flexible and general, including different situations, scenarios, and populations. The characteristics presented and the delimitations established are suggestive of a scoping review. The methods describe the study endpoints but do not describe about possible subgroup analyses. It is important to describe heterogeneity criteria and possible sensitivity analysis. I suggest you add embase as a database. Did the authors use any automation tools? Didn't the authors use a reference manager? Was the selection of studies only manual? I suggest that the authors describe the assumptions adopted for cases of missing or unclear information in more detail. In table S2, I suggest that next to each database the number of articles found is added. In the title, I suggest you indicate the date of the last search. The results were adequately presented. The discussion does not provide a general interpretation of the results in the context of other evidence. Most of the discussion concerns the implications of the results for practice and policy, but before that, reflections and possible interpretations need to be discussed and debated in greater depth. The limitations of the study need to be more clearly described. Analysis of the confidence of the evidence was not analyzed.

Reviewer #4: in this SRMA, Domnich and Calabrò have provided a pooled estimate on the occurrence of RSV infections in Italy. Notably, only in recent years has Italy implemented an official, nation-wide surveillance for RSV infections: as a consequence, until recently the actual occurrence of RSV in Italy was a big "question mark", and only some local studies (most of them designed for tracing and tracking the occurrence of several respiratory pathogens, and not only and not specifically RSV) have tried to provide some estimates on RSV.

All of the aforementioned potential limits have been specifically explained, discussed and reported across the main text.

More specifically, Authors have discussed in very cautionary terms the pooled estimates they've provided, mostly because of the background quality of data they did recollect. From the point of view of the present reviewer, this is the main issue that Authors should address before the full acceptance of the present study.

The ROB assessment was performed by means of the JBI checklist and, according to the Table S5, the overall quality of recollected papers wast mostly appropriate. On the contrary, the main text suggests (at least from my understanding) that the overall quality of the papers was mostly affected by several shortcomings, particularly when dealing with the potential selection of patients and diagnosis of RSV infection. A specifically designed section of results or better in discussion section should be therefore implemented in order to reconcile the substantially good results from ROB analysis and the shortcomings that were specifically addressed by Authors.

Another issue that should be addressed by Authors is stressing more extensively the heterogeneity in reported outcomes from recollected studies: as suggested by Table 1, cases that were included in the analyses were quite heterogeneous (ARI, ILI, SARI, etc), and even the timeframe of the study (in vs. out of season study) may have either inflated or reduced the eventual prevalence of RSV. Again, Authors have implemented an appropriate methodological approach to address this shortcoming, but a more extensive and detailed discussion of this specific shortcoming is requested (either through a new specifically designed limits section or as a section after rows 390 and following).

6. PLOS authors have the option to publish the peer review history of their article (what does this mean?). If published, this will include your full peer review and any attached files.

Reviewer #1: No

Reviewer #2: No

Reviewer #3: No

Reviewer #4: No

---

## [Author Response · Author response to Decision Letter 0]

7 Feb 2024

Editor’s comments

Comment: 1. Please ensure that your manuscript meets PLOS ONE's style requirements, including those for file naming. The PLOS ONE style templates can be found at 

Reply: The style has been changed accordingly.

Comment: 2. Thank you for stating the following in the Competing Interests section: 

"AD and GEC declare no competing interests." Please confirm that this does not alter your adherence to all PLOS ONE policies on sharing data and materials, by including the following statement: ""This does not alter our adherence to PLOS ONE policies on sharing data and materials.” (as detailed online in our guide for authors http://journals.plos.org/plosone/s/competing-interests). If there are restrictions on sharing of data and/or materials, please state these. Please note that we cannot proceed with consideration of your article until this information has been declared. Please include your updated Competing Interests statement in your cover letter; we will change the online submission form on your behalf.

Reply: The statement suggested has been added in both the manuscript and cover latter.

Comment: 3. We note that this manuscript is a systematic review or meta-analysis; our author guidelines therefore require that you use PRISMA guidance to help improve reporting quality of this type of study. Please upload copies of the completed PRISMA checklist as Supporting Information with a file name “PRISMA checklist”.

Reply: The PRISMA checklist file was uploaded and is currently named “S1 Table PRISMA checklist”

Reviewer #1

Comment: This is a comprehensive assessment of the respiratory burden of RSV in italian adults. The methodology is robust. The article is well written.

Reply: Thank you for your interest in our paper. Your comment below has been addressed.

Comment: I would add in the discussion and in the title that this work focused on the respiratory burden of RSV (and not the well-known cardio-vascular complications or other middle/long term complications such as re hospitalizations, dependancy etc...)

Reply: Thank you for this comment. Our review was focused on both respiratory and extra-respiratory complications of RSV and related hospitalizations. However, no studies on the extra-respiratory consequences as well as other important indicators of the natural history of RSV infection have been available. We have now clearly indicated this data lack in the Methods and Discussion. Related suggestions for future research have been also made.

Reviewer #2

Comment: Authors of this manuscript looked into epidemiology and burden of RSV infection in Italian adults. Their research was carried our as a systematic literature review and met analysis. Overall, this works shows a very clear structure and follows the generally accepted rules for such research. The field of RSV epidemiological studies is pretty broad and with varying data and output quality. It is a strength of this work that authors focused on relevant outcomes and parameters to be extracted from analysed research. Data is well described and conclusions are well supported by their findings. As the authors state, this article provides a comprehensive assessment of the RSV epidemiology and BOD in the Italian population. Shortcomings of the existing literature are in a way limiting the strength of the conclusions of the paper, which is - of course - not a fault of the submitting researcher.

Reply: Thank you for your interest in our paper. We have now further deepened and substantiated our discussion. A particular emphasis has been made for limitations of the current evidence.

Reviewer #3

Comment: Dear all, The systematic review presented is relevant and has important contributions to the literature; however, some items must be reviewed when conducting the study. I suggest checking Prisma and faithfully following the checklist established as a guide for quality systematic reviews. Page MJ, McKenzie JE, Bossuyt PM, Boutron I, Hoffmann TC, Mulrow CD, et al. The PRISMA 2020 statement: an updated guideline for reporting systematic reviews. BMJ 2021;372:n71. doi: 10.1136/bmj.n71

Reply: Thank you for your interest in our paper. All your comments have been addressed. We have further checked our adherence to the PRISMA 2020 checklist. 

Comment: In the abstract: - There is no clear statement of the main research question.

Reply: The research question has been now clearly stated.

Comment: - There is no specification of inclusion and exclusion criteria, nor are the databases used to identify the studies described. 

Reply: Both eligibility criteria and database searched have been now reported.

Comment: - There is no description of the last date on which the literature searches were carried out. 

Reply: As required, this has been stated.

Comment: - There is no detailed description of the methods used to assess the risk of bias in the included studies.

Reply: As required, this has been added.

Comment: - Authors must include the registration number of the review protocol in the study abstract. 

Reply: Information on the protocol registration has been added.

Comment: - There is no adequate synthesis of the main results. 

Reply: We have now assured the all the principal results are summarized in the abstract. Please note, according to the PLOS ONE’s guidelines, the abstract must not exceed 300 words.

Comment: - The conclusion of the abstract talks about public policies, assessment of health technologies and vaccines. This is not the conclusion of a systematic review. Nor is the main finding of this study added to a pertinent conclusion related to what the results of the study contribute to the literature based on their possible generalizations.

Reply: Conclusions provided in the abstract have been now completely revised. Conclusions on public policies have been now removed, as suggested.

Comment: In the article: The authors claim to use Prisma and follow the guidelines of Joanna Briggs' institute; however, important details highlighted by these standards need to be added and clearly described, allowing for replicability of the review and transparency in its methodological conduct.

Reply: We have now assured that all the reporting standards have been met.

Comment: The authors use the acronym CoCoPop (condition, context, and population) which is usually used for scoping reviews. For cross-sectional studies, the acronym PECO (pollution, exposure, comparison, and outcome) would be ideal.

Reply: The CoCoPop framework has been specifically designed for systematic reviews and/or meta-analyses of incidence and prevalence studies (https://doi.org/10.1186%2Fs12874-017-0468-4, https://doi.org/10.1097/xeb.0000000000000054). Conversely, the PECO framework has been specifically developed (https://doi.org/10.1016/j.envint.2018.07.015) for environmental and occupational health research (pollution, etc.).

Comment: What is the guiding question of the study? What are the prevalence, incidence, and risk factors of respiratory syncytial virus in Italian adults? This is not clear.

Reply: Our research question was ““In a typical winter season, how many Italian adults contract RSV and how many of them develop complications, being hospitalized and die?” This has been now clearly reported.

Comment: The eligibility criteria established by the authors are quite flexible and general, including different situations, scenarios, and populations. The characteristics presented and the delimitations established are suggestive of a scoping review.

Reply: Thank you for this comment. According to the landmark paper by Munn et al. (https://doi.org/10.1186/s12874-018-0611-x), systematic reviews are indicated to “Produce statements to guide decision-making”. This was indeed our ultimate goal, also in view of a recent availability of adult RSV vaccines, for which no recommendations have been made so far. Conversely, “…scoping reviews are an ideal tool to determine the scope or coverage of a body of literature on a given topic and give clear indication of the volume of literature and studies available”. In our review, the study population, condition of interest and context are all rather specific and were set in order to address our specific research question. The only flexible characteristic is represented by the RSV-related outcomes. This is, however, inherent to respiratory infections and vaccine-preventable diseases in general; in this regard, the available Cochrane reviews on vaccines cover a plethora of different outcomes. It is also true that the manuscript title could be changed into “…A systematic scoping review and meta-analysis”. We, however, believe that such a title would be ambiguous.

Comment: The methods describe the study endpoints but do not describe about possible subgroup analyses. It is important to describe heterogeneity criteria and possible sensitivity analysis.

Reply: In the Methods (Data analysis, §2), we have clearly reported that “To investigate the sources of heterogeneity across studies, both subgroup and meta-regression analyses were performed. In particular, the pre-specified subgroup analysis was performed by age-group (working-age and older adults), setting (outpatient, inpatient and mixed) and study period in relation to the COVID-19 pandemic (before the 2020/2021 season and after the 2020/2021 season). The meta-regression modeling was then conducted to examine the influence of study characteristics on the RSV-related endpoints.” The corresponding results have been presented in Figures 2 and 3, as well as in supplementary materials.

Comment: I suggest you add embase as a database. 

Reply: Unfortunately, our institutions did not allow to access EMBASE. This latter, however, has more of a focus on drug and chemicals, which is not our case. In any case, we have now added this potential study limitation.

Comment: Did the authors use any automation tools? Didn't the authors use a reference manager? Was the selection of studies only manual?

Reply: We used Zotero and MS Excel. This has been now clearly reported.

Comment: I suggest that the authors describe the assumptions adopted for cases of missing or unclear information in more detail.

Reply: Missing, unclear or presented only in graphical form data on relevant numerators and/or denominators were handled as follows. First, the corresponding author was contacted for clarification. In case of no reply, these data were imputed from the available percentages and/or by extracting data from figures using the WebPlotDigitizer software. This has been clearly stated (see section “Data extraction and abstraction”, last paragraph).

Comment: In table S2, I suggest that next to each database the number of articles found is added. In the title, I suggest you indicate the date of the last search.

Reply: As suggested, the date of the last search and the number of articles found hasve abben added to Table S2.

Comment: The results were adequately presented.

Reply: Thank you. No changes required.

Comment: The discussion does not provide a general interpretation of the results in the context of other evidence. Most of the discussion concerns the implications of the results for practice and policy, but before that, reflections and possible interpretations need to be discussed and debated in greater depth. The limitations of the study need to be more clearly described. Analysis of the confidence of the evidence was not analyzed.

Reply: Thank you for this comment. We have now revised the entire Discussion section. Review limitations have been deepened and contextualized (paragraphs 6 and 7). The entire paragraphs 3-5 of the Discussion are dedicated to the context of other available evidence. An emphasis on the observed heterogeneity has been made and possible causes of the latter have been discussed. More general policy reflections have been significantly reduced.

Reviewer #4

Comment: In this SRMA, Domnich and Calabrò have provided a pooled estimate on the occurrence of RSV infections in Italy. Notably, only in recent years has Italy implemented an official, nation-wide surveillance for RSV infections: as a consequence, until recently the actual occurrence of RSV in Italy was a big "question mark", and only some local studies (most of them designed for tracing and tracking the occurrence of several respiratory pathogens, and not only and not specifically RSV) have tried to provide some estimates on RSV. All of the aforementioned potential limits have been specifically explained, discussed and reported across the main text. More specifically, Authors have discussed in very cautionary terms the pooled estimates they've provided, mostly because of the background quality of data they did recollect. From the point of view of the present reviewer, this is the main issue that Authors should address before the full acceptance of the present study.

Reply: Thank you for your interest in our study. Your comments have been now addressed.

Comment: The ROB assessment was performed by means of the JBI checklist and, according to the Table S5, the overall quality of recollected papers was mostly appropriate. On the contrary, the main text suggests (at least from my understanding) that the overall quality of the papers was mostly affected by several shortcomings, particularly when dealing with the potential selection of patients and diagnosis of RSV infection. A specifically designed section of results or better in discussion section should be therefore implemented in order to reconcile the substantially good results from ROB analysis and the shortcomings that were specifically addressed by Authors.

Reply: Thank you for this comment. This apparent discrepancy is linked to the absence of a universally recognized RSV case definition. Therefore, reviewers’ judgement on the risk of bias due to sample frame is rather subjective. In the present review, we intentionally did not downgrade studies that used ILI- or influenza-specific syndromic definitions (also because studies conducted before the COVID-19 pandemic were a bit “forced” to conduct study within the established influenza surveillance platforms. Although it is believed that ILI-based surveillance underestimates RSV detection rates, some studies found no difference between ILI- and ARI-based RSV case definitions in terms of sensitivity and specificity. We have now clearly indicated and discussed this issue among the review-level limitations (§7 of the Discussion).

Comment: Another issue that should be addressed by Authors is stressing more extensively the heterogeneity in reported outcomes from recollected studies: as suggested by Table 1, cases that were included in the analyses were quite heterogeneous (ARI, ILI, SARI, etc), and even the timeframe of the study (in vs. out of season study) may have either inflated or reduced the eventual prevalence of RSV. Again, Authors have implemented an appropriate methodological approach to address this shortcoming, but a more extensive and detailed discussion of this specific shortcoming is requested (either through a new specifically designed limits section or as a section after rows 390 and following).

Reply: We completely agree with your observation on the heterogeneity of RSV detection rates obtained in our random-effects models. We have now explicitly stated this limitation and discussed on its possible sources in a specific paragraph of the limitations related to the current evidence (see §6 of the Discussion).

---

## [Decision Letter · Decision Letter 1]

19 Feb 2024

Epidemiology and burden of respiratory syncytial virus in Italian adults: A systematic review and meta-analysis

PONE-D-24-00880R1

Dear Dr. Domnich,

We’re pleased to inform you that your manuscript has been judged scientifically suitable for publication and will be formally accepted for publication once it meets all outstanding technical requirements.

Kind regards,

Victor Daniel Miron

Academic Editor

PLOS ONE

Reviewers' comments:

Reviewer's Responses to Questions

**Comments to the Author**

1. If the authors have adequately addressed your comments raised in a previous round of review and you feel that this manuscript is now acceptable for publication, you may indicate that here to bypass the “Comments to the Author” section, enter your conflict of interest statement in the “Confidential to Editor” section, and submit your "Accept" recommendation.

Reviewer #3: All comments have been addressed

Reviewer #4: All comments have been addressed

2. Is the manuscript technically sound, and do the data support the conclusions?

Reviewer #3: Yes

Reviewer #4: Yes

3. Has the statistical analysis been performed appropriately and rigorously? 

Reviewer #3: Yes

Reviewer #4: Yes

4. Have the authors made all data underlying the findings in their manuscript fully available?

Reviewer #3: Yes

Reviewer #4: Yes

5. Is the manuscript presented in an intelligible fashion and written in standard English?

Reviewer #3: Yes

Reviewer #4: Yes

6. Review Comments to the Author

Reviewer #3: (No Response)

Reviewer #4: Estimated Authors,

all my concerns and doubts about the original design of the paper have been properly addressed.

Therefore, I'm endorsing the eventual acceptance of this study.

7. PLOS authors have the option to publish the peer review history of their article (what does this mean?). If published, this will include your full peer review and any attached files.

Reviewer #3: No

Reviewer #4: **Yes: **Matteo Riccò

---

## [Editor Report · Acceptance letter]

23 Feb 2024

PONE-D-24-00880R1 

PLOS ONE

Dear Dr. Domnich, 

I'm pleased to inform you that your manuscript has been deemed suitable for publication in PLOS ONE. Congratulations! Your manuscript is now being handed over to our production team.

Kind regards, 

on behalf of

Dr. Victor Daniel Miron 

Academic Editor

PLOS ONE